# What Drives Bitcoin? An Approach from Continuous Local Transfer Entropy and Deep Learning Classification Models

**DOI:** 10.3390/e23121582

**Published:** 2021-11-26

**Authors:** Andrés García-Medina, Toan Luu Duc Huynh

**Affiliations:** 1Unidad Monterrey, Centro de Investigación en Matemáticas, A.C. Av. Alianza Centro 502, PIIT, Apodaca 66628, Mexico; 2Consejo Nacional de Ciencia y Tecnología, Av. Insurgentes Sur 1582, Col. Crédito Constructor, Ciudad de México 03940, Mexico; 3WHU—Otto Beisheim School of Management, 56179 Düsseldorf, Germany; toanhld@ueh.edu.vn; 4UEH Institute of Innovation (UII), University of Economics Ho Chi Minh City, Ho Chi Minh City 70000, Vietnam; 5IPAG Business School, 75006 Paris, France

**Keywords:** local transfer entropy, long-short-term-memory, Bitcoin

## Abstract

Bitcoin has attracted attention from different market participants due to unpredictable price patterns. Sometimes, the price has exhibited big jumps. Bitcoin prices have also had extreme, unexpected crashes. We test the predictive power of a wide range of determinants on bitcoins’ price direction under the continuous transfer entropy approach as a feature selection criterion. Accordingly, the statistically significant assets in the sense of permutation test on the nearest neighbour estimation of local transfer entropy are used as features or explanatory variables in a deep learning classification model to predict the price direction of bitcoin. The proposed variable selection do not find significative the explanatory power of NASDAQ and Tesla. Under different scenarios and metrics, the best results are obtained using the significant drivers during the pandemic as validation. In the test, the accuracy increased in the post-pandemic scenario of July 2020 to January 2021 without drivers. In other words, our results indicate that in times of high volatility, Bitcoin seems to self-regulate and does not need additional drivers to improve the accuracy of the price direction.

## 1. Introduction

Currently, there is tremendous interest in determining the dynamics and direction of the price of Bitcoin due to its unique characteristics, such as its decentralization, transparency, anonymity, and speed in carrying out international transactions. Recently, these characteristics have attracted the attention of both institutional and retail investors. Thanks to technological developments, investor trading strategies are benefited by digital platforms; therefore, market participants are more likely to digest and create information for this market. Of special interest is its decentralized character, since its value is not determined by a central bank but, essentially, only by supply and demand, recovering the ideal of a free market economy. At the same time, it is accessible to all sectors of society, which breaks down geographic and particular barriers for investors. The fact that there are a finite number of coins and the cost of mining new coins grows exponentially has suggested to some specialists that it may be a good instrument for preserving value. That is, unlike fiat money, Bitcoin cannot be arbitrarily issued, so its value is not affected by the excessive issuance of currency that central banks currently follow, or by low interest rates as a strategy to control inflation. In other words, it has been recently suggested that bitcoin is a safe-haven asset or store of value, having a role similar to that once played by gold and other metals.

The study of cryptocurrencies and bitcoin has been approached from different perspectives and research areas. It has been addressed from the point of view of financial economics, econometrics, data science, and more recently by econophysics. In these approaches, various methodologies and mathematical techniques have been utilised to understand different aspects of these new financial instruments. These topics range from systemic risk, the spillover effect, autoscaling properties, collective patterns, price formation, and forecasting in general. Remarkable work in the line of multiscale analysis of cryptocurrency markets can be found in [1]. However, this paper is motivated by using the econphysics approach, incorporated with rigorous control variables to predict Bitcoin price patterns. We would like to offer a comprehensive review of the determinants of Bitcoin prices. The first pillar can be defined as sentiment and social media content. While Bitcoin is widely considered a digital financial asset, investors pay attention to this largest market capitalization by searching its name. Therefore, the strand of literature on Google search volume has become popular for capturing investor attention [2]. Concomitantly, not only peer-to-peer sentiment (individual Twitter accounts or fear from investors) [3,4] but also influential accounts (the U.S. President, media companies) [5,6,7] significantly contribute to Bitcoin price movement. Given the greatest debate on whether Bitcoin should act as a hedging, diversifying or safe-haven instrument, Bitcoin exhibits a mixture of investing features. More interestingly, uncertain shocks might cause changes in both supply and demand in Bitcoin circulation, implying a change in its prices [8]. Thus, the diverse stylized facts of Bitcoin, including heteroskedasticity and long memory, require uncertainty to be controlled in the model. While uncertainties represent the amount of risk (compensated by the Bitcoin returns) [9], our model also includes the price of risk, named the ‘risk aversion index’ [10]. These two concepts (amount of risk and the price of risk) demonstrate discount rate factors in the time variation of any financial market [11]. In summary, the appearance of these determinants could capture the dynamics of the cryptocurrency market. Since cryptocurrency is a newly emerging market, the level of dependence in the market structure is likely higher than that in other markets [12]. Furthermore, the contagion risk and the connectedness among these cryptocurrencies could be considered the risk premium for expected returns [13,14]. More importantly, this market can be driven by small market capitalization, implying vulnerability of the market [15]. Hence, our model should contain alternative coins (altcoins) to capture their movements in the context of Bitcoin price changes. Finally, investors might consider the list of these following assets as alternative investment, precious metals being the first named. They are not only substitute assets [16] but also predictive factors (for instance, gold and platinum) [17], which additionally include commodity markets (such as crude oil [18,19], exchange rate [20], equity market [21]), and Tesla’s owner [22]). In summary, there are voluminous determinants of Bitcoin prices. In the scope of this study, we focus on the predictability of our model, especially the inclusion of social media content, representing the high popularity of information, on the Bitcoin market. However, the more control variables there are, the higher the accuracy of prediction. Our model thus may be a useful tool by combining the huge predictive factors for training and forecasting the response dynamics of Bitcoin to other relevant information.

This study approaches Bitcoin from the framework of behavioural and financial economics using an approach from econophysics and data science. In this sense, it seeks to understand the speculative character and the possibilities of arbitrage through a model that includes investor attention and the effect of the news, among other factors. For this, we will use a causality method originally proposed by Schreiber [23], and we will use the information as characteristics of a deep learning model. The current literature only focuses on specific sentiment indicators (such as Twitter users [3] or the number of tweets [24,25]), and our study crawled the original text from influential Twitter social media users (such as the President of United States, CEO of Tesla, and well-known organizations such as the United Nations and BBC Breaking News). Then, we processed language analyses to construct the predictive factor for Bitcoin prices. Therefore, our model incorporates a new perspective on Bitcoin’s drivers.

In this direction, the work of [26] uses the effective transfer entropy as an additional feature to predict the direction of U.S. stock prices under different machine learning approaches. However, the approximation is discrete and based on averages. Furthermore, the employed metrics are not exhaustive to determine the predictive power of the models. In a similar vein, the authors of [27] perform a comparative analysis of machine learning methods for the problem of measuring asset risk premiums. Nevertheless, they do not take into account recurrent neural network models or additional nontraditional features. Furthermore, an alternative approach to study the main drivers of Bitcoin is discussed in [28], where the author explores wavelet coherence to examine the time and frequency domains between short- and long-term interactions. In the same vein, the recent studies employed the correlation networks and vector error correction models to explain the price prediction and exchange spillovers [29,30]. Of course, Bitcoin prediction is more likely to have sentimental and ‘noise’ factors differing from stock prediction.

On the other hand, there are methodologies to explain machine learning results known as eXplainable Artificial Intelligence (XAI). Among these, two of the most popular are Local Interpretable Model Agnostic [31] and Shapley Additive Explanation (SHAP) [32]. Both techniques are based on disturbing the model locally. The former assumes a linear model to obtain the score of the characteristics in terms of the importance of making predictions; the latter uses game theory concepts to find the best feature fitting in terms of predictive gain. In [33] these techniques are extended to include temporal dependencies and demonstrate the need to develop XAI techniques applicable to time series. In [34,35] is proposed an XAI method applicable to credit risk. In a similar vein, the authors of [36] mention the difficulty of estimating out-of-sample behavior in stress scenarios. An interesting work is [37], where it is considered a gradient boosting decision trees approximation to predict the drops of the S&P 500 markets using a large number of characteristics. The authors claim that retaining a small and carefully selected amount of features improves the learning model results.

However, as mentioned in the cornerstone work [31] it is not possible to explain a highly non-linear model through local perturbations. That is, there is a high instability derived from the characteristics of the inherent dynamical system. In addition, the examples of the articles mentioned above run in most cases in seconds or minutes. Therefore, the LIME and SHAP methods are appropriate mainly for machine learning models or simple deep learning scenarios [38]. In this spirit, it is not practical to follow the traditional XAI approach, given the computational demand derived from the number of hyperparameters and configurations to be implemented. However, our proposal to use transfer entropy in the variable selection process can be considered an alternative strategy to XAI. In particular, of interest for highly non-linear dependency conditions, such as bitcoin dynamics.

Our study embodied a wide range of Bitcoin’s drivers from alternative investment, economic policy uncertainty, investor attention, and so on. However, social media is our main contribution to predictive factors. Specifically, we study the effect that a set of Twitter accounts belonging to politicians and millionaires has on the behaviour of Bitcoin’s price direction. In this work, the statistically significant drivers of Bitcoin are detected in the sense of the continuous estimation of local transfer entropy (local TE) through nearest neighbours and permutation tests. The proposed methodology deals with non-Gaussian data and nonlinear dependencies in the problem of variable selection and forecasting. One main aim is to quantify the effects of investor attention and social media on Bitcoin in the context of behavioural finance. Another aim is to apply classification metrics to indicate the effects of including or not the statistically significant features in an LSTM’s classification problem.

The next Section 2 introduce the local transfer entropy, the nearest neighbour estimation technique, the deep learning forecasting models, and the classification metrics. Section 3 describes the data and their main descriptive characteristics. Section 4 presents and highlights the main results. Finally, Section 5 highlights the implications of the results, and future work is proposed.

## 2. Materials and Methods

### 2.1. Transfer Entropy

Transfer Entropy (TE) [23] measures the flow of information from system *Y* over system *X* in a nonsymmetric way. Denote the sequences of states of systems *X*, *Y* in the following way: $x_{i}=x\left(i\right)$ and $y_{i}=y\left(i\right),i=1,\ldots ,N$. The idea is to model the signals or time series as Markov systems and incorporate the temporal dependencies by considering the states $x_{i}$ and $y_{i}$ to predict the next state $x_{i+1}$. If there is no deviation from the generalized Markov property $p\left(x_{i+1}|x_{i},y_{i}\right)=p\left(x_{i+1}|x_{i}\right)$, then *Y* has no influence on *X*. Hence, TE is derived using the last idea and defined as
(1)$$T_{Y\to X}\left(k,l\right)=\sum p\left(x_{i+1},x_{i}^{\left(k\right)},y_{i}^{\left(l\right)}\right)\log\frac{p(x_{i+1}|x_{i}^{\left(k\right)},y_{i}^{\left(l\right)})}{p(x_{i+1}|x_{i}^{\left(k\right)})},$$
where $x_{i}^{\left(k\right)}=\left(x_{i},\ldots ,x_{i-k+1}\right)$ and $y_{i}^{\left(l\right)}=\left(y_{i},\ldots ,y_{i-l+1}\right)$.

TE can be thought of as a global average or expected value of a local transfer entropy at each observation [39]
(2)$$T_{Y\to X}\left(k,l\right)=\langle \log\frac{p(x_{i+1}|x_{i}^{\left(k\right)},y_{i}^{\left(l\right)})}{p(x_{i+1}x_{i}^{\left(k\right)})}\rangle$$
The main characteristic of the local version of TE is to be measured at each time *n* for each destination element *X* in the system and each causal information source *Y* of the destination. It can be either positive or negative for a specific event set $\left(x_{i+1},x_{i}^{\left(k\right)},y_{i}^{\left(l\right)}\right)$, which gives the opportunity to have a measure of informativeness or noninformativeness at each point of a pair of time series.

On the other hand, there exist several approximations to estimate the probability transition distributions involved in TE expression. Nevertheless, there is not a perfect estimator. It is generally impossible to minimize both the variance and the bias at the same time. Then, it is important to choose the one that best suits the characteristics of the data under study. That is the reason finding good estimators is an open research area [40]. This study followed the Kraskov-Stögbauer-Grassberger) KSG estimator [41], which focused on small samples for continuous distributions. Their approach is based on nearest neighbours. Although obtaining insight into this estimator is not easy, we will try it in the following.

Let $X=(x_{1},x_{2},\ldots ,x_{d})$ now denote a d-dimensional continuous random variable whose probability density function is defined as $p:R^{d}\to R$. The continuous or differential Shannon entropy is defined as
(3)$$H\left(X\right)=-\int_{R^{d}}p\left(X\right)\log p\left(X\right)dX$$
The KSG estimator aims to use similar length scales for K-nearest-neighbour distance in different spaces, as in the joint space to reduce the bias [42].

To obtain the explicit expression of the differential entropy under the KSG estimator, consider *N* i.i.d. samples $\chi ={\left\{X\left(i\right)\right\}}_{i=1}^{N}$, drawn from p(X). Beneath the assumption that $\epsilon_{i,K}$ is twice the (maximum norm) distance to the k-th nearest neighbour of X(i), the differential entropy can be estimated as
(4)$${\hat{H}}_{KSG,K}\left(X\right)\equiv \psi \left(N\right)-\psi \left(K\right)+\frac{d}{N}\sum_{i=1}^{N}\log\epsilon_{i,K},$$
where ψ is known as the digamma function and can be defined as the derivative of the logarithm of the gamma function Γ(x)
(5)$$\psi \left(K\right)=\frac{1}{\Gamma (K)}\frac{d\Gamma (K)}{dK}$$

The parameter *K* defines the size of the neighbourhood to use in the local density estimation. It is a free parameter, but there exists a trade-off between using a smaller or larger value of *K*. The former approach should be more accurate, but the latter reduces the variance of the estimate. For further intuition, Figure 1 graphically shows the mechanism for choosing the nearest neighbours at K=3.

The KSG estimator of TE can be derived based on the previous estimation of the differential entropy. Yet, in most cases, as analysed in this work, no analytic distribution is known. Hence, the distribution of $T_{Y_{s}\to X}\left(k,l\right)$ must be computed empirically, where $Y_{s}$ denotes the surrogate time series of *Y*. This is done by a resampling method, creating a large number of surrogate time-series pairs $\left\{Y_{s},X\right\}$ by shuffling (for permutations or redrawing for bootstrapping) the samples of Y. In particular, the distribution of $T_{Y_{s}\to X}\left(k,l\right)$ is computed by permutation, under which surrogates must preserve $p(x_{n+1}|x_{n})$ but not $p(x_{n+1}|x_{n},y_{n})$.

### 2.2. Deep Learning Models

We can think of artificial neural networks (ANNs) as a mathematical model whose operation is inspired by the activity and interactions between neuronal cells due to their electrochemical signals. The main advantages of ANNs are their non-parametric and nonlinear characteristics. The essential ingredients of an ANN are the neurons that receive an input vector $x_{i}$, and through the point product with a vector of weights *w*, generate an output via the activation function g(·):(6)$$f\left(x_{i}\right)=g\left(x_{u}\cdot w\right)+b,$$
where *b* is a trend to be estimated during the training process. The basic procedure is the following. The first layer of neurons or input layer receives each of the elements of the input vector $x_{i}$ and transmits them to the second (hidden) layer. The next hidden layers calculate their output values or signals and transmit them as an input vector to the next layer until reaching the last layer or output layer, which generates an estimation for an output vector.

Further developments of ANNs have brought recurrent neural networks (RNNs), which have connections in the neurons or units of the hidden layers to themselves and are more appropriate to capture temporal dependencies and therefore are better models for time series forecasting problems. Instead of neurons, the composition of an RNN includes a unit, an input vector $x_{t}$, and an output signal or value $h_{t}$. The unit is designed with a recurring connection. This property induces a feedback loop, which sends a recurrent signal to the unit as the observations in the training data set are analysed. In the internal process, backpropagation is performed to obtain the optimal weights. Unfortunately, backpropagation is sensitive to long-range dependencies. The involved gradients face the problem of vanishing or exploding. Long-short-term memory (LSTM) models were introduced by Hochreiter and Schmidhuber [43] to avoid these problems. The fundamental difference is that LSTM units are provided with memory cells and gates to store and forget unnecessary information.

The final ANNs we need to discuss are convolutional neural networks (CNNs). They can be thought of as a kind of ANN that uses a high number of identical copies of the same neuron. This allows the network to express computationally large models while keeping the number of parameters small. Usually, in the construction of these types of ANNs, a max-pooling layer is included to capture the largest value over small blocks or patches in each feature map of previous layers. It is common that CNN and pooling layers are followed by a dense fully connected layer that interprets the extracted features. Then, the standard approach is to use a flattened layer between the CNN layers and the dense layer to reduce the feature maps to a single one-dimensional vector [44].

### 2.3. Classification Metrics

In classification problems, we have the predicted class and the actual class. The possible scenarios under a classification prediction are given by the confusion matrix. They are true positive (TP), true negative (TN), false positive (FP), and false negative (FN). Based on these quantities, it is possible to define the following classification metrics:Accuracy = $\frac{TP+TN}{TP+TN+FP+FN}$Sensitivity, recall or true positive rate (TPR) $=\frac{TP}{TP+FN}$Specificity, selectivity or true negative rate (TNR) $=\frac{TN}{TN+FP}$Precision or Positive Predictive Value (PPV) $=\frac{TP}{TP+FP}$False Omission Rate (FOR) $=\frac{FN}{FN+TN}$Balanced Accuracy (BA) $=\frac{TPR+TNR}{2}$F1 score $=2\frac{PPV\times TPR}{PPV+TPR}$.

The most complex measure is the area under the curve (AUC) of the receiver operating characteristic (ROC), where it expresses the pair $\left(TPR_{\tau },1-TNR_{\tau }\right)$ for different thresholds τ. Contrary to the other metrics, the AUC of the ROC is a quality measure that evaluates all the operational points of the model. A model with the aforementioned metric equal to 0.5 is considered a random model. Then, a value significantly higher than 0.5 is considered a model with predictive power, with a value of 1 the upper bound of this quantity.

## 3. Data

An important part of the work is the acquisition and preprocessing of data. We focus on the period of time from 1 January 2017 to 9 January 2021 at a daily frequency for a total of n=1470 observations. As a priority, we consider the variables listed in Table 1 as potential drivers of the price direction of Bitcoin (BTC). Investor attention is considered Google Trends with the *query = “Bitcoin”*. Additionally, the number of mentions is properly scaled to make comparisons between days of different months because by default, Google Trends weighs the values by a monthly factor. Then, the log return of the resulting time series is calculated.

The social media data are collected from the Twitter API (https://developer.twitter.com/en/docs/twitter-api, accessed on 15 January 2021). Nevertheless, the API of Twitter only enables downloading the latest 3200 tweets of a public profile, which generally was not enough to cover the period of study. Then, the dataset has been completed with the freely available repository of https://polititweet.org/ (accessed on 15 January 2021). In this way, the collected number of tweets was 21,336, 22,808, 24,702, 11,140, and 26,169 for each of the profiles listed on Table 1 in the social media type, respectively. The textual data of each tweet in the collected dataset are transformed to a sentiment polarity score through the VADER lexicon [45]. Then, the scores are aggregated daily for each profile. The resulting daily time series have missing values due to the inactivity of the users, and then a third-order spline is considered before calculating their differences. The last is to stationarize the polarity time series. It is important to remember that Donald Trump’s account was blocked on 8 January 2021, so it was also necessary to impute the last value to have series of the same length.

The economic policy uncertainty index is a Twitter-based uncertainty index (Twitter-EPU). The creators of the index used the Twitter API to extract tweets containing keywords related to uncertainty (“uncertain”, “uncertainly”, “uncertainties”, “uncertainty”) and economy (“economic”, “economical”, “economically”, “economics”, “economies”, “economist”, “economists”, “economy”). Then, we use the index consisting of the total number of daily tweets containing inflections of the words uncertainty and economy (Please consult https://www.policyuncertainty.com/twitter_uncert.html for further details of the index, accessed on 15 January 2021). The risk aversion category considers the financial proxy to risk aversion and economic uncertainty proposed as a utility-based aversion coefficient [10]. A remarkable feature of the index is that in early 2020, it reacted more strongly to the new COVID-19 infectious cases than did a standard uncertainty proxy.

As complementary drivers, it includes a set of highly capitalized cryptocurrencies and a heterogeneous portfolio of financial indices. Specifically, Ethereum (ETH), Litecoin (LTC), Ripple (XRP), Dogecoin (DOGE), and the stable coin TETHER are included from yahoo finance (https://finance.yahoo.com/, accessed on 15 January 2021). The components of the heterogeneous portfolio are listed in Table 1, which takes into account the Chicago Board Options Exchange’s CBOE Volatility Index (VIX). This last information was extracted from Bloomberg (https://www.bloomberg.com/, accessed on 15 January 2021). It is important to point out that risk aversion and the financial indices do not have information that corresponds to weekends. The imputation method to obtain a complete database consisted of repeated Friday values as a proxy for Saturday and Sunday. Then, the log return of the resulting time series is calculated. This last transformation was also made for Twitter-EPU and cryptocurrencies. The complete dataset can be found in the Appendix A.

Usually, the econophysics and data science approaches share the perspective of observing data first and then modelling the phenomena of interest. In this spirit, and with the intention of gaining intuition on the problem, the standardized time series (target and potential drivers), as well as the cumulative return of the selected cryptocurrencies and financial assets are plotted in Figure 2 and Figure 3. The former figure shows high volatility in almost all the studied time series around March 2020, which might be due to the declaration of the pandemic by the World Health Organization (WHO) and the consequent fall of the worlds main stock markets. The latter figure exhibits the overall best cumulative gains for BTC, ETH, LTC, XRP, DOGE, and Tesla. It is worth noting that the only asset with a comparable profit to that of the cryptocurrencies is Tesla, which reaches high cumulative returns starting at the end of 2019 and increases its uptrend immediately after the announcement of the worldwide health emergency.

Furthermore, Figure 4 shows the heatmap of the correlation matrix of the preprocessed dataset. We can observe the formation of certain clusters, such as cryptocurrencies, metals, energy, and financial indices, which tells us about the heterogeneity of the data. It should also be noted that the VIX volatility index is anti-correlated with most of the variables.

Additionally, the main statistical descriptors of the data are presented in Table 2. The first column is the variable’s names or tickers. The subsequent columns represent the mean, standard deviation, skewness, kurtosis, Jarque Bera test (JB), and the associated *p* value of the test for each variable, i.e., target, and potential drivers. Basically, none of the time series passes the test of normality distribution, and most of them present a high kurtosis, which is indicative of heavy tail behaviour. Finally, stationarity was checked in the sense of Dickey-Fuller and the Phillips-Perron unit root tests, where all variables pass both tests.

## 4. Results

### 4.1. Variable Selection

The observed characteristics of the data in the previous section justify the use of a non-parametric approach to determine the explainable features to be employed in the predictive classification model. Therefore, the variable selection procedure consisted of applying the continuous transfer entropy from each driver to Bitcoin using the KSG estimation. Figure 5 shows the average transfer entropy when varying the Markov order k,l and neighbour parameter *K* from one to ten for a total of 1000 different estimations by each driver. The higher the intensity of the colour, the higher the average transfer entropy (measured in nats). The grey cases do not transfer information to BTC. In other words, these cases do not show a statistically significant flow of information, where the permutation test is applied to construct 100 surrogate measurements under the null hypothesis of no directed relationship between the given variables.

The tuple of parameters {k,l,K} that give the highest average transfer entropy from each potential driver to BTC are considered optimal, and the associated local TE is kept as a feature in the classification model of Bitcoin’s price direction. Figure 6 shows the local TE from each statistically significant driver to BTC at the optimal parameter tuple {k,l,K}. Note that the set of local TE time series is limited to 23 features. Consequently, the set of originally proposed potential drivers is reduced from 25 to 23. Surprisingly, NASDAQ and Tesla do not send significative information to BTC for any value of {k,l,K} in the grid of the 1000 different configurations. The variations are smooth on *K*, but not on the Markov order k,l. It is also notorious the negligible amounts of the flow of information at k=l=1.

### 4.2. Bitcoin’s Price Direction

The task of detecting Bitcoin’s price direction was done through a deep learning approach. The first step consisted of splitting the data into training, validation, and test datasets. The chosen training period runs from 1 January 2017 to 4 January 2020, or 75% of the original entire period of time, and is characterized as a prepandemic scenario. The validation dataset is restricted to the period from 5 January 2020 to 11 July 2020, or 13% of the original data, and is considered the pandemic scenario. The test dataset involves the postpandemic scenario from 12 July 2020 to 9 January 2021 and contains 12% of the complete dataset. Deep learning forecasting requires transforming the original data into a supervised data set. Here, samples of 74 historical days and a one-step prediction horizon are given to the model to obtain a supervised training dataset, with the first dimension being a power of two, which is important for the hyperparameter selection of the batch dimension. Specifically, the sample dimensions are 1024, 114, and 107 for training, validation, and testing, respectively. Because we are interested in predicting the direction of BTC, the time series are not demeaned and instead are only scaled by their variance when feeding the deep learning models. An important piece in a deep learning model is the selection of the activation function. In this work, the rectified linear unit (ReLU) was selected for the hidden layers. Then, for the output layer, the sigmoid function is chosen because we are dealing with a classification problem. In addition, an essential ingredient is the selection of the stochastic gradient descent method. Here, Adam optimization is selected based on adaptive estimation of the first- and second-order moments. In particular, we used version [46] to search for the long-term memory of past gradients to improve the convergence of the optimizer.

There exist several hyperparameters to take into account when modelling a classification problem under a deep learning approach. These hyperparameters must be calibrated on the training and validation datasets to obtain reliable results on the test dataset. The usual procedure to set them is via a grid search. Nevertheless, deeper networks with more computational power are necessary to obtain the optimal values in a reasonable amount of time. To avoid excessive time demands, we vary the most crucial parameters in a small grid and apply some heuristics when required. The number of epochs, is selected under the early stopping procedure. Another crucial hyperparameter is the batch, or the number of samples to work through before updating the internal weight of the model. For this parameter the selected grid was {32,64,128,256}. Additionally, we consider the initial learning rates at which the optimizer starts the algorithm, which were {0.001,0.0001}. As an additional method of regularization, the effect of dropping between consecutive layers is added. This value can take values from 0 to 1. Our grid for this hyperparameter is {0.3,0.5,0.7}. Finally, because of the stochastic nature of the deep learning models, it is necessary to run several realizations and work with averages. We repeat the hyperparameter selection with ten different random seeds for robustness. The covered scenarios are the following: *univariate* (S1), where bitcoin is self-driven; *all features* (S2), where all the potential drivers listed in Table 1 are included as features of the model; *significative features* (S3), only statistically significant drivers under the KSG transfer entropy approach are considered as features; *local TE*, only the local TE of the statistically significant drivers are included as a feature; and finally the *significative features + local TE* (S5) scenario, which combines scenarios (S3) and (S4). Finally, five different designs have been proposed for the architectures of the neural networks, which are denoted as *deep LSTM* (D1), *wide LSTM* (D2), *deep bidirectional LSTM* (D3), *wide bidirectional LSTM* (D4), and *CNN* (D5). The specific designs and diagrams of these architectures are displayed in Figure 7. In total, 6000 configurations or models were executed, which included the grid search for the optimal hyperparameters, the different scenarios and architectures, and the realizations on different seeds to avoid biases due to the stochastic nature of the considered machine learning models.

The computation was done in a workstation with the following characteristics: Alienware Aurora R7, Ubuntu 20.10, Processor i9-9900X 8 cores, 16 logic, 64 GB RAM, Dual NVIDIA RTX 2080 ti, 3TB HHD. On this equipment, the computational demand extends the execution to nearly 60 h of computation. Table 3 and Table 4 present the main results for the validation and test datasets, respectively. Table 2 explicitly states the best value for the dropout, learning rate (LR), and batch hyperparameters. In both tables, the hashtag (#) column indicates the number of times the specific scenario gives the best score for the different metrics considered so far. Hence, the architecture design D3 for case S3 yields the highest number of metrics with the best scores in the validation dataset. In contrast, in the test dataset, the highest number of metrics with the best scores correspond to design D2 for case S1. Nevertheless, design D5 from case S5 is close in the sense of the # value, where it presents the best AUC and PPV scores. An important point to keep in mind is that only during the validation stage we find models with an AUC greater than 0.6, so this metric does not give evidence of predictive power in the testing stage.

In a robustness discussion, we would like to compare our predictive feature with the existing approaches. While the current studies look at the conventional approach of econometrics [29,30], our study sheds light on the deep learning method. Accordingly, we had two samples (training sample and test group). Therefore, it allows us to validate our findings with different periods. The unique, comparable study that we have found in the area of learning models is due to [26]. However, they only show the results for two accuracy metrics when predicting the direction of the US markets. Even so, barely the metrics exceed the value of 0.6, and it is not clear if they are considering a test set.

## 5. Discussion

We start from descriptive statistics as a first approach to intuitively grasp the complex nature of Bitcoin, as well as its proposed heterogeneous drivers. As expected, the variables did not satisfy the normality assumption and presented high kurtosis, highlighting the need to use non-parametric and nonlinear analyses.

The KSG estimation of TE found a consistent flow of information from the potential drivers to Bitcoin through the considered range of K nearest neighbours. Even when, in principle, the variance of the estimate decreases with *K*, the results obtained with K=1 do not change abruptly for larger values. In fact, the variation in the structure of the TE matrix for different Markov orders k,l is more notorious. Additionally, attention must be paid to the evidence about the order k=l=1 through values near zero. Practitioners usually assume this scenario under Gaussian estimations. A precaution must then be made about the memory parameters of Markov, at least when working with the KSG estimation. The associated local TE does not show any particular pattern beyond high volatility, reaching values of four nats when the average is below 0.1. Thus, volatility might be a better proxy for price fluctuations in future studies.

In terms of intuitive explanations, we found that the drivers of Bitcoin might not truly capture its returns in distressed periods. Although we expected to witness that the predictive power of these determinants might play an important role across time horizons, it turns out that the prediction model of Bitcoin relies on a choice of a specific period. Thus, our findings also confirm the momentum effect that exists in this market [47]. Due to the momentum effect, the timing of market booms could not truly be supported much for further analysis by our models. In regard to our main social media hypothesis, the popularity of Bitcoin content still exists as the predictive component in the model. More noticeably, our study highlights that Bitcoin prices can be driven by momentum on social media [24]. However, the selection of training and testing periods should be cautious with the boom and burst of this cryptocurrency. Apparently, while the fundamental value of Bitcoin is still debatable [48], using behavioural determinants could have some merits in predicting Bitcoin. Thus, we believe that media content would support the predictability of Bitcoin prices alongside other financial indicators. Concomitantly, after clustering these factors, we found that the results seem better able to provide insights into Bitcoin’s drivers.

On the other hand, the forecasting of Bitcoin’s price direction improves in the validation set but not for all metrics in the test dataset when including significant drivers or local TE as a feature. Nonetheless, the last assertion relies on the number of metrics with the best scores. Although the test dataset having the best performance corresponds to the *deep bidirectional LSTM* (D3) for the scenario *univariate* (S3), this case only beat three of the eight metrics. The other five metrics are outperformed by scenarios including *significative features* (S3) and *significative features + local TE* (S5). Furthermore, the second-best performances are tied with two of the eight metrics with leading values. Interestingly, the last case shows the best predictive power on the CNN model using significant features as well as local TE indicators (D5–S5). In particular, it outperforms the AUC and PPV overall, yet AUC is in the border of a random model. To delve into the explainable aspect, a future work will seek to apply the Shapley-Lorentz decomposition proposed in [49,50]. There the authors develop a global methodology, which can be associated with a generalization of AUC-ROC.

Moreover, it is important to note that the selected test period is atypical in the sense of a bull period for Bitcoin as a result of the turbulence generated by the COVID-19 public health emergency; this might induce safe haven behaviour related to this asset and increase its price and capitalization. This atypical behaviour opens the door to propose future work to model Bitcoin by the self-exciting process of the Hawkes model during times of great turbulence.

We would like to end by emphasizing that we were not exhaustive in modelling classification forecasting. In contrast, our intention was to exemplify the effect of including the significant features and local TE indicators under different configurations of a deep learning model through a variety of classification metrics. Two methodological contributions to highlight are the use of nontraditional indicators such as market sentiment, as well as a continuous estimation of the local TE as a tool to determine additional drivers in the classification model. Finally, the models presented here are easily adaptable to high-frequency data because they are non-parametric and nonlinear in nature.

## Figures and Tables

**Figure 1 entropy-23-01582-f001:**
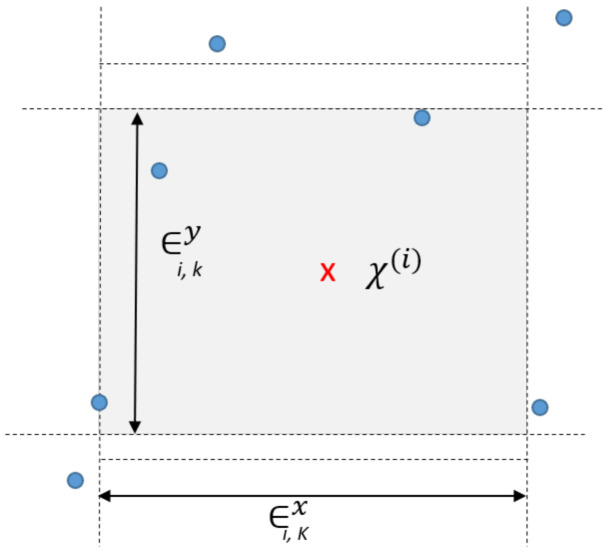
Graphical representation of nearest-neighbors selection. At a given sample point, X(i), the max-norm rectangle contains the K=3 nearest-neighbors.

**Figure 2 entropy-23-01582-f002:**
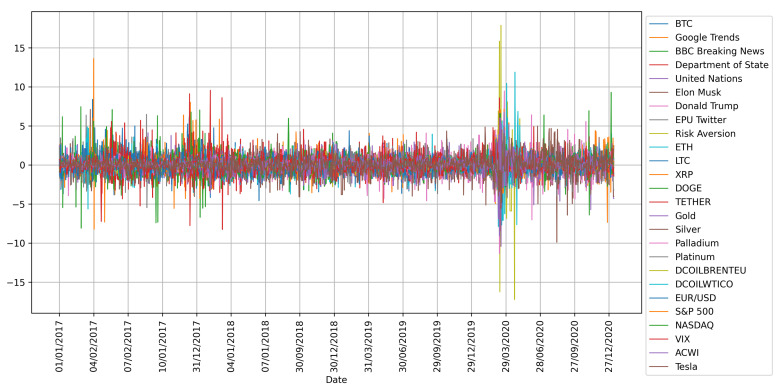
Standardized time series after preprocessing, as explained in the main text.

**Figure 3 entropy-23-01582-f003:**
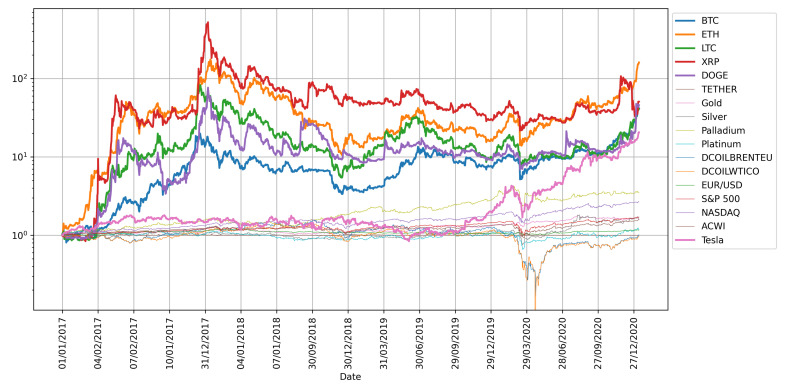
Cumulative returns of the selected cryptocurrencies and financial assets. The scale is logarithmic in the y-axis and starts in one to be financially interpreted as the gains.

**Figure 4 entropy-23-01582-f004:**
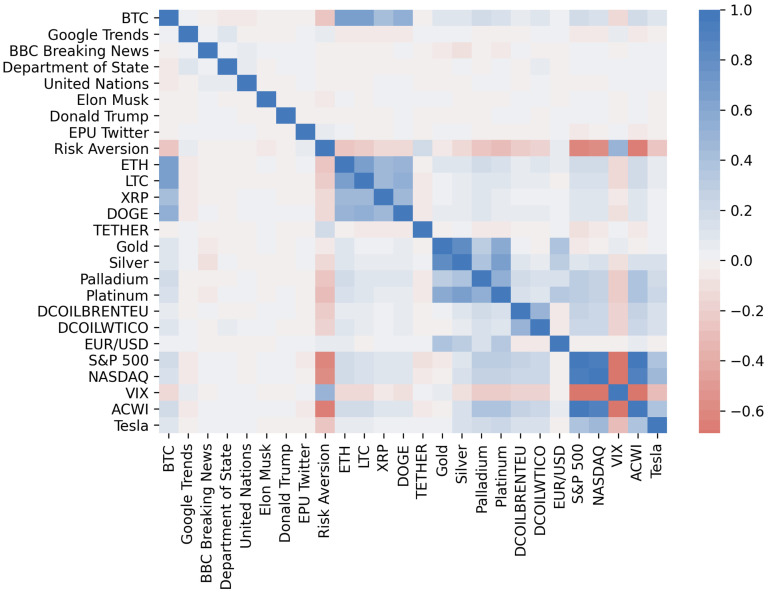
Correlation matrix of the preprocessed time series.

**Figure 5 entropy-23-01582-f005:**
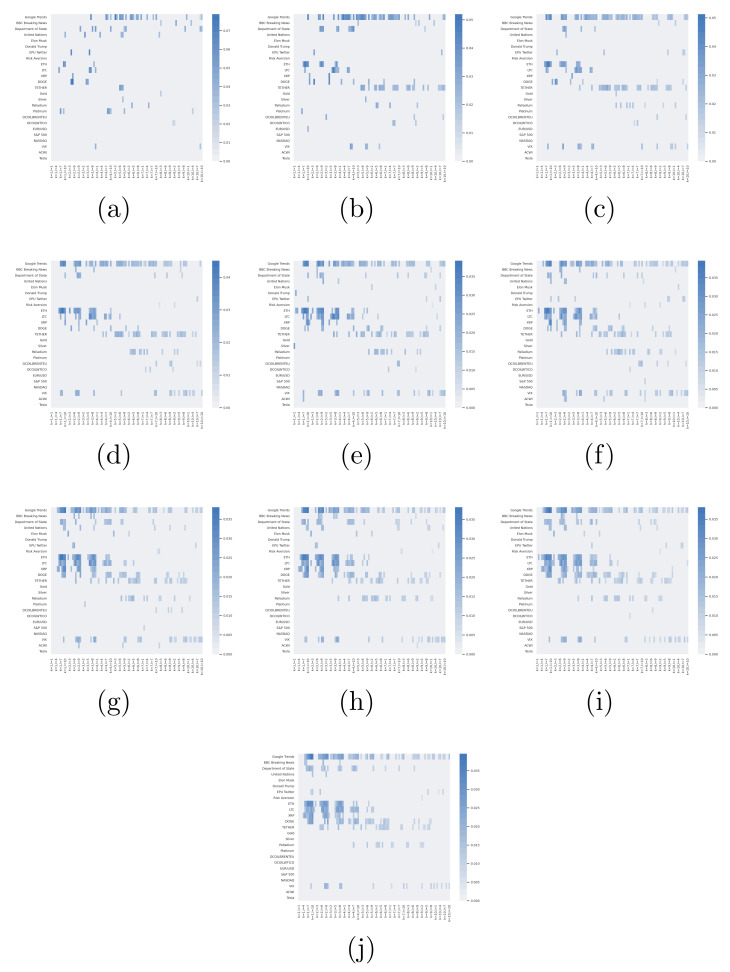
Average transfer entropy from each potential driver to BTC. The y-axis indicates the driver, and the x-axis indicates the Markov order pair k,l of the source and target. From (**a**) to (**j**), nearest neighbours *K* run from one to ten, respectively.

**Figure 6 entropy-23-01582-f006:**
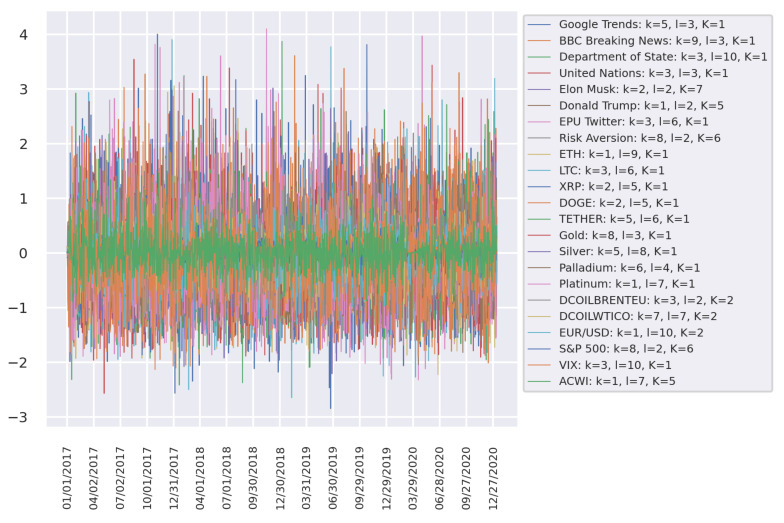
Local TE of the highest significant average values on the tuple {k,l,K}. NASDAQ and Tesla are omitted because they do not send significative information to BTC for any considered value on the grid of the tuple {k,l,K}.

**Figure 7 entropy-23-01582-f007:**
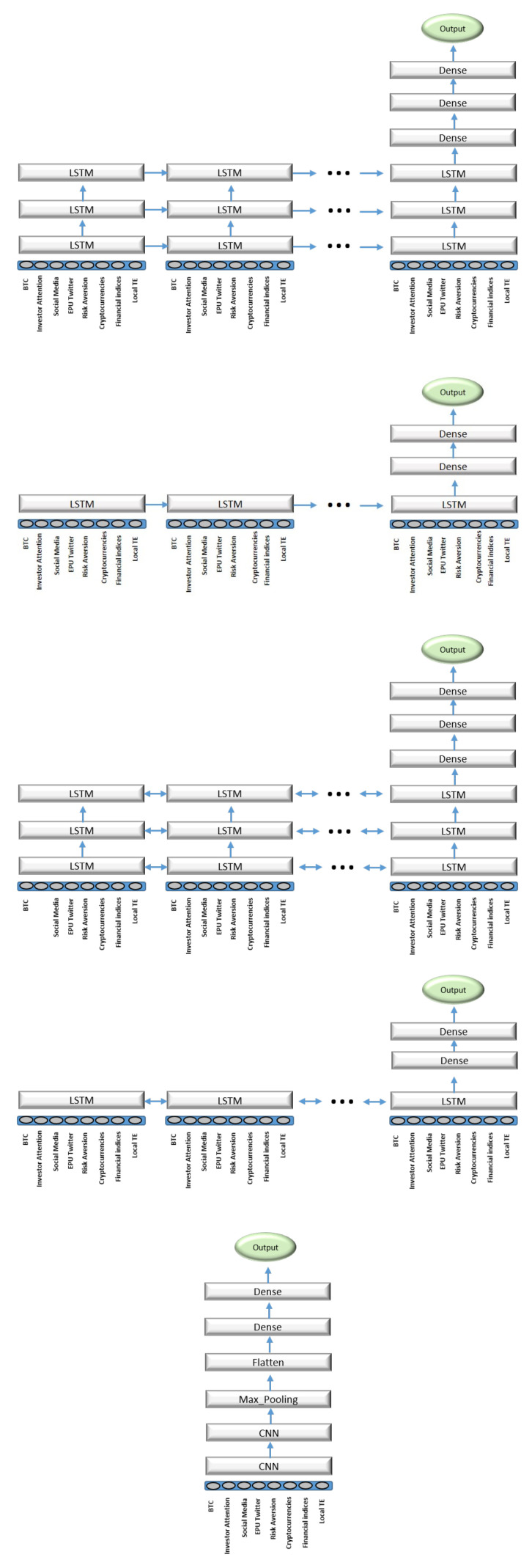
From (**top**) to (**bottom**): D1, D2, D3, D4, and D5.

**Table 1 entropy-23-01582-t001:** Type of driver and variable name.

Type	Variables
Investor attention	Google Trends
Social media	BBC Breaking News
	Department of State
	United Nations
	Elon Musk
	Donald Trump
Twitter-EPU	Twitter-based Uncertainty Index
Risk Aversion	Financial Proxy to Risk Aversion and Economic Uncertainty
Cryptocurrencies	ETH
	LTC
	XRP
	DOGE
	TETHER
Financial indices	Gold
	Silver
	Palladium
	Platinum
	DCOILBRENTEU
	DCOILWTICO
	EUR/USD
	S&P 500
	NASDAQ
	VIX
	ACWI
	Tesla

**Table 2 entropy-23-01582-t002:** The symbols **, and *** denote the significance at the 5%, and 1% levels, respectively.

Variable	Mean	Std. Dev.	Skewness	Kurtosis	JB	*p*-Value
BTC	0.0025	0.0424	−0.8934	12.7470	10,073.5034	***
Google Trends	0.0018	0.1915	0.2611	6.8510	2868.6001	***
BBC Breaking News	0.0007	3.5295	−0.1789	15.5376	14,686.4748	***
Department of State	−0.0013	4.0610	0.0941	8.4698	4362.1014	***
United Nations	0.0007	3.2689	0.1748	0.2747	11.9228	**
Elon Musk	0.0035	1.8877	0.0672	3.8630	906.9805	***
Donald Trump	0.0001	4.8294	0.0461	5.2434	1670.4686	***
Twitter−EPU	0.0009	0.3001	0.3049	3.6071	812.4777	***
Risk Aversion	0.0001	0.0726	3.7594	165.2232	1,664,075.5884	***
ETH	0.0034	0.0566	−0.3991	9.7009	5759.1438	***
LTC	0.0025	0.0606	0.6919	10.5404	6870.4947	***
XRP	0.0027	0.0753	2.2903	36.2405	81,162.9262	***
DOGE	0.0026	0.0669	1.2312	15.0342	14,113.2759	***
TETHER	0.0000	0.0062	0.3255	20.0952	24,581.8501	***
Gold	0.0006	0.0082	−0.6595	5.5761	1995.0834	***
Silver	0.0005	0.0164	−1.1304	13.0841	10,720.4362	***
Palladium	0.0015	0.0197	−0.9198	20.5312	25,840.0775	***
Platinum	0.0004	0.0145	−0.9068	10.7274	7196.3125	***
DCOILBRENTEU	−0.0004	0.0374	−3.1455	81.2272	403,755.7220	***
DCOILWTICO	0.0006	0.0358	0.7362	38.4244	89,931.3161	***
EUR/USD	0.0002	0.0042	0.0336	0.8999	49.0930	***
S&P 500	0.0006	0.0125	−0.5714	20.5446	25,746.7436	***
NASDAQ	0.0008	0.0145	−0.3601	11.7771	8463.6169	***
VIX	−0.0061	0.0810	1.4165	8.4537	4833.8826	***
ACWI	0.0006	0.0115	−1.1415	20.4837	25,833.5682	***
Tesla	0.0017	0.0371	−0.3730	5.5089	1877.5034	***

**Table 3 entropy-23-01582-t003:** Classification metrics on the validation dataset.

Design	Case	Dropout	LR	Batch	Acc	AUC	TPR	TNR	PPV	FOR	BA	F1	#
D1	S1	0.3	0.001	32	57.11	0.5388	84.75	25.28	56.63	40.97	55.02	67.89	
	S2	0.3	0.001	128	57.28	0.5391	80.33	30.75	57.18	42.40	55.54	66.80	
	S3	0.7	0.001	128	58.07	0.5379	74.43	39.25	58.51	42.86	56.84	65.51	
	S4	0.3	0.001	256	57.98	0.5304	75.41	37.92	58.30	42.74	56.67	65.76	
	S5	0.3	0.001	256	57.19	0.5100	87.54	22.26	56.45	39.18	54.90	**68.64**	
D2	S1	0.3	0.001	64	59.82	0.5444	81.97	34.34	58.96	**37.67**	58.15	68.59	
	S2	0.3	0.001	32	61.14	0.5909	65.57	56.04	63.19	41.42	60.81	64.36	
	S3	0.3	0.001	128	**62.28**	**0.6062**	62.95	**61.51**	**65.31**	40.94	**62.23**	64.11	5
	S4	0.5	0.0001	32	55.44	0.4964	75.08	32.83	56.27	46.63	53.96	64.33	
	S5	0.7	0.001	32	58.07	0.5706	63.77	51.51	60.22	44.74	57.64	61.94	
D3	S1	0.3	0.001	128	56.23	0.4865	**88.52**	19.06	55.73	40.94	53.79	68.40	
	S2	0.3	0.001	64	59.65	0.5816	68.69	49.25	60.90	42.26	58.97	64.56	
	S3	0.3	0.001	128	60.09	0.5619	76.72	40.94	59.92	39.55	58.83	67.29	
	S4	0.3	0.001	32	58.16	0.5350	79.18	33.96	57.98	41.37	56.57	66.94	
	S5	0.3	0.001	256	59.47	0.5702	68.69	48.87	60.72	42.44	58.78	64.46	
D4	S1	0.5	0.001	32	57.28	0.5276	80.16	30.94	57.19	42.46	55.55	66.76	
	S2	0.7	0.001	128	58.68	0.5447	66.23	50.00	60.39	43.74	58.11	63.17	
	S3	0.7	0.001	64	58.25	0.5468	64.26	51.32	60.31	44.49	57.79	62.22	
	S4	0.5	0.001	256	57.11	0.5092	78.36	32.64	57.25	43.28	55.50	66.16	
	S5	0.7	0.0001	32	57.11	0.5328	70.33	41.89	58.21	44.91	56.11	63.70	
D5	S1	0.7	0.001	128	60.09	0.5834	72.13	46.23	60.69	40.96	59.18	65.92	
	S2	0.3	0.001	64	60.00	0.5683	67.70	51.13	61.46	42.09	59.42	64.43	
	S3	0.5	0.001	32	59.39	0.5648	68.03	49.43	60.76	42.67	58.73	64.19	
	S4	0.5	0.001	32	59.12	0.5572	75.57	40.19	59.25	41.16	57.88	66.43	
	S5	0.3	0.001	128	60.79	0.5825	70.33	49.81	61.73	40.67	60.07	65.75	

**Table 4 entropy-23-01582-t004:** Classification metrics on the test dataset.

Design	Case	Acc	AUC	TPR	TNR	PPV	FOR	BA	F1	#
D1	S1	64.30	0.4981	90.99	11.67	67.01	60.38	51.33	77.18	
	S2	56.82	0.4946	74.08	22.78	65.42	69.17	48.43	69.48	
	S3	61.78	0.5188	79.58	26.67	68.15	**60.17**	**53.12**	73.42	2
	S4	51.96	0.4898	60.70	34.72	64.71	69.06	47.71	62.65	
	S5	60.47	0.4842	83.66	14.72	65.93	68.64	49.19	73.74	
D2	S1	60.75	0.4786	85.92	11.11	65.59	71.43	48.51	74.39	
	S2	52.06	0.4870	56.48	43.33	66.28	66.45	49.91	60.99	
	S3	53.46	0.4997	56.76	46.94	67.85	64.50	51.85	61.81	
	S4	55.70	0.4794	70.56	26.39	65.40	68.75	48.48	67.89	
	S5	50.93	0.4806	55.49	41.94	65.34	67.67	48.72	60.02	
D3	S1	**65.05**	0.5072	**95.21**	5.56	66.54	62.96	50.38	**78.33**	3
	S2	55.70	0.5248	63.38	40.56	67.77	64.04	51.97	65.50	
	S3	57.38	0.5176	67.32	37.78	68.09	63.04	52.55	67.71	
	S4	51.40	0.5051	52.96	48.33	66.90	65.75	50.65	59.12	
	S5	54.21	0.5094	60.56	41.67	67.19	65.12	51.12	63.70	
D4	S1	61.21	0.4831	86.48	11.39	65.81	70.07	48.93	74.74	
	S2	48.13	0.4718	48.17	48.06	64.65	68.02	48.11	55.21	
	S3	47.20	0.4771	43.24	**55.00**	65.46	67.05	49.12	52.08	
	S4	45.98	0.4359	50.99	36.11	61.15	72.80	43.55	55.61	
	S5	51.96	0.4743	55.49	45.00	66.55	66.11	50.25	60.52	
D5	S1	58.04	0.5017	78.59	17.50	65.26	70.70	48.05	71.31	
	S2	55.23	0.4942	62.39	41.11	67.63	64.34	51.75	64.91	
	S3	54.21	0.4994	63.10	36.67	66.27	66.50	49.88	64.65	
	S4	54.49	0.5269	62.39	38.89	66.82	65.60	50.64	64.53	
	S5	55.79	**0.5316**	61.83	43.89	**68.49**	63.17	52.86	64.99	2

## Data Availability

The data is available as Appendix A.

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
