# Peer review of "What Drives Bitcoin? An Approach from Continuous Local Transfer Entropy and Deep Learning Classification Models"

_entropy, 2021, doi:10.3390/e23121582_

Round 1

Reviewer 1 Report

The paper concerns a relevant topic: the prediction of bitcoin prices, using an ample set of regressors, that include twitter sentiment data. The employed methodologies (transfer entropy, deep learning) are not new but the authors fine tune them to the specific problem. 

The paper is of interest but, to be considered for acceptance, it should be improved in some directions, as follows:

  1. literature  review: the authors mention, but not cite, related works on biticoin price prediction. I suggest to cite at least some of them, for example the following and also  the references therein which refer to bitcoin price determination and discovery:

    Giudici, P., Polinesi, G. (2021).  Crypto price discovery through correlation networks. Annals of operations research, 2021, 299 (1-2), 443-457.

    Giudici, P., Pagnottoni, P. (2020) Vector error correction models to measure connectedness of bitcoin exchange markets. Applied stochastic models in business and industry, 36 (1), 95-109.

2) Accuracy. The performance of the proposed methods, in terms of AUROC, is not high, as it never exceeds 0.60. The authors wrongly state that a value of AUROC greater than 0.5 indicates a good model. That is just the value of a pure random model!

3) Robustness. The authors should compare their methods, in predictive performance, against methods employed in other papers, including the ones cited above. 

4) Explainability. The authors should try to explain the results of their deep learning models, using model agnostic techniques such as SHAP and LIME. At least they should quote papers that deal with explainable AI methods in pricing and risk management

5) Conciseness. the methodological section could be shortened as the methods are known. Similarly, the results section should be shortened and the main results highlighted.

Overall, the paper deals with an important topic, and I look forward to a revised version that takes my comments into account

Author Response

Reply: We appreciate the observations and comments made on the work, which ideas undoubtedly have helped to improve the quality of the manuscript. The following summarizes the main quantitative changes made to this new version of the manuscript:

  • The number of references has been increased from 40 to 47.
  • The figure related to the LSTM unit structure has been removed.

Reviewer #1

The paper concerns a relevant topic: the prediction of bitcoin prices, using an ample set of regressors, that include twitter sentiment data. The employed methodologies (transfer entropy, deep learning) are not new but the authors fine tune them to the specific problem.

The paper is of interest but, to be considered for acceptance, it should be improved in some directions, as follows:

  1. Literature review: the authors mention, but not cite, related works on bitcoin price prediction. I suggest to cite at least some of them, for example the following and also  the references therein which refer to bitcoin price determination and discovery:

Giudici, P., Polinesi, G. (2021). Crypto price discovery through correlation networks. Annals of operations research, 2021, 299 (1-2), 443-457.

Giudici, P., Pagnottoni, P. (2020) Vector error correction models to measure connectedness of bitcoin exchange markets. Applied stochastic models in business and industry, 36 (1), 95-109.

Reply: Thank you for your thoughtful comments. We elaborated and explained the price prediction based on the two suggested references. 

2) Accuracy. The performance of the proposed methods, in terms of AUROC, is not high, as it never exceeds 0.60. The authors wrongly state that a value of AUROC greater than 0.5 indicates a good model. That is just the value of a pure random model!

Reply: The text where this metric is presented has been corrected to appropriately describe the meaning of the values:

...Contrary to the other metrics, the AUC of the ROC is a quality measure that evaluates all the operational points of the model. A model with the aforementioned metric equal to 0.5 is considered a random model. Then, a value significantly higher than 0.5 is considered a model with predictive power, with a value of 1 the upper bound of this quantity…”

We have also added the observation of the behavior of the AUC in the results section:

...An important point to keep in mind is that only during the validation stage we found models with an AUC greater than 0.6, so this metric does not give evidence of predictive power in the testing stage…”

3) Robustness. The authors should compare their methods, in predictive performance, against methods employed in other papers, including the ones cited above.

Reply: We also elaborated the advantages of our method in the last paragraph of discussion. Thank you for your thoughtful comments. 

4) Explainability. The authors should try to explain the results of their deep learning models, using model agnostic techniques such as SHAP and LIME. At least they should quote papers that deal with explainable AI methods in pricing and risk management.

Reply: Undoubtedly, they are state-of-the-art techniques that will be sought to be applied in a more suitable experimental design. In this problem the computation time of the 6000 configurations was approximately 60 hours on a workstation. So disrupting the system locally would considerably increase computational complexity, so the result might not be even ready within the assigned review time.

In the introduction are discussed previous works in the context of the XAI models, and the approach followed in this work is justified as a consequence of some weaknesses of these techniques at the current stage of their development. Moreover, in the result section it is described the specifications of the workstation and also the computation time consumed is declared.

5) Conciseness. the methodological section could be shortened as the methods are known. Similarly, the results section should be shortened and the main results highlighted.

To attend this point the following changes were made:

  • The explicit derivation of the local entropy transfer was removed
  • Some expressions related to the KSG estimate of entropy transfer were removed.
  • The formulas of the LSTM model and their graphic representation were removed.
  • The subsection Data was separated as proper section for a better presentation of the results, and to highlight in a more concise way the most relevant findings.
  • We removed non essential information on the remaining section of results.

Overall, the paper deals with an important topic, and I look forward to a revised version that takes my comments into account

Reviewer 2 Report

no comments

Author Response

Reply: We appreciate the observations and comments made on the work, which ideas undoubtedly have helped to improve the quality of the manuscript. The following summarizes the main quantitative changes made to this new version of the manuscript:

  • The number of references has been increased from 40 to 47.
  • The figure related to the LSTM unit structure has been removed.

Round 2

Reviewer 1 Report

The authors took into account most of my suggestions. I would further emphasize the importance of using explainable AI methods in cryptocurrency analysis as described in the paper Giudici and Raffinetti, 2021, "Shapley Lorenz Explainable Artificial Intelligence", expert systems with applications and some of the references therein

Apart from this remark, which I hope the authors can take into account, I have no other objections

Author Response

Thanks for suggesting this reference. We have added it in the discussion section as future work along with another of the same authors. We believe this is very relevant topic closely related to our work, so we are looking forward to apply those ideas in an upcoming study.

Round 3

Reviewer 1 Report

The paper can be accepted